# In vivo proteomic mapping through GFP-directed proximity-dependent biotin labelling in zebrafish

Zherui Xiong[1], Harriet P Lo[1], Kerrie-Ann McMahon[1], Nick Martel[1], Alun Jones[1], Michelle M Hill[2], Robert G Parton[1,3]*, Thomas E Hall[1]*

[1]Institute for Molecular Bioscience, The University of Queensland, Brisbane, Australia; [2]QIMR Berghofer Medical Research Institute, Herston, Australia; [3]Centre for Microscopy and Microanalysis, The University of Queensland, Brisbane, Australia

**Abstract** Protein interaction networks are crucial for complex cellular processes. However, the elucidation of protein interactions occurring within highly specialised cells and tissues is challenging. Here, we describe the development, and application, of a new method for proximity-dependent biotin labelling in whole zebrafish. Using a conditionally stabilised GFP-binding nanobody to target a biotin ligase to GFP-labelled proteins of interest, we show tissue-specific proteomic profiling using existing GFP-tagged transgenic zebrafish lines. We demonstrate the applicability of this approach, termed BLITZ (Biotin Labelling In Tagged Zebrafish), in diverse cell types such as neurons and vascular endothelial cells. We applied this methodology to identify interactors of caveolar coat protein, cavins, in skeletal muscle. Using this system, we defined specific interaction networks within in vivo muscle cells for the closely related but functionally distinct Cavin4 and Cavin1 proteins.

*For correspondence:
r.parton@imb.uq.edu.au (RGP);
thomas.hall@imb.uq.edu.au (TEH)

Competing interests: The authors declare that no competing interests exist.

## Introduction

The understanding of the biological functions of a protein requires detailed knowledge of the molecules with which it interacts. However, robust elucidation of interacting proteins, including not only strong direct protein-protein interactions, but also weak, transient or indirect interactions is challenging. Proximity-dependent biotin labelling (BioID) using genetically engineered biotin ligases has emerged as a novel approach for studying protein-protein interactions and the subcellular proteome in living cells (*Roux et al., 2012*; *Kim et al., 2016*; *Branon et al., 2018*; *Ramanathan et al., 2018*). When fused to a protein of interest (POI) and expressed in cells, the promiscuous biotin ligases covalently attach biotin to all proteins within a 10 nm radius, which can be subsequently isolated by streptavidin purification and identified by mass spectrometry. Compared with traditional affinity purification with protein-specific antibodies or affinity purification tags, the BioID method has the advantage of being able to capture weak and transient interactions. In addition, unlike conventional methods such as affinity purification, where stringent extraction conditions may disrupt protein-protein interactions, the BioID method does not require proteins to be isolated in their native state. Therefore, harsh protein extraction and stringent wash conditions can be applied, which can improve solubilisation of membrane proteins and reduce false positives (*Varnaité and MacNeill, 2016*; *Gingras et al., 2019*).

The BioID method has been widely applied in cell biology to study protein-protein interactions in cultured cells, providing valuable information for building protein interaction networks. However, the reductionist in vitro applications described to date, while powerful in their own right, lack the complexity and context to address phenomena that can only be modelled in vivo, for example the differentiation of specialised cell types such as those found in muscle, the nervous system, and

vasculature. The most recent generation of biotin ligases has been applied in vivo in invertebrate models; flies (*Drosophila melanogaster*) and worms (*Caenorhabditis elegans*) as well as plants (*Arabidopsis* and *Nicotiana benthamiana*) (*Branon et al., 2018*; *Mair et al., 2019*; *Zhang et al., 2019*). Until now, however, the applicability of BioID has been limited by the necessity to genetically tag each POI directly with a biotin ligase and generate transgenic organisms. Here, we describe a more versatile approach to the in vivo application of BioID in a vertebrate model organism, the zebrafish. Instead of directly fusing the biotin ligase to a POI, we developed a modular system for GFP-directed proteomic mapping by combining BioID with a GFP-binding nanobody (*Hamers-Casterman et al., 1993*; *Rothbauer et al., 2008*; *Tang et al., 2015*; *Ariotti et al., 2015a*). This system couples the power of the BioID system with the ability to use existing GFP-tagged transgenic zebrafish lines for proteomic mapping between different tissues and/or different proteins. We demonstrate the application of this system in screening for proteins associated with the caveolar cast proteins, Cavin1 and Cavin4, in differentiated skeletal muscle which has, to date, been difficult to achieve in culture. These analyses reveal proteins and pathways that are both overlapping and specific to Cavin1 and Cavin4.

## Results

### Proximity biotinylation in live zebrafish embryos

We first tested the ability of a number of biotin ligases to catalyse protein biotinylation in live zebrafish embryos. Initial attempts using BirA* or BioID2 biotin ligases in vivo in zebrafish were unsuccessful and resulted in no detectable biotinylation in zebrafish embryos as assessed by streptavidin western blotting (*Figure 1—figure supplement 1*). In recent years, the new bioID biotin ligases, BASU, TurboID, and miniTurbo, have been developed and showed greatly improved catalytic efficiency and enhanced proximity labelling in cultured cells (*Branon et al., 2018*; *Ramanathan et al., 2018*). We therefore tested their ability to catalyse protein biotinylation in live zebrafish embryos. Untagged, cytoplasmically localised biotin ligases were transiently expressed in zebrafish embryos by RNA injection, and the level of protein biotinylation was assessed using streptavidin immunoblot analysis (experimental regimen illustrated in *Figure 1A*; *Ramanathan et al., 2018*; *Branon et al., 2018*). The biotin ligases were fused to an EGFP tag for selection of transgene expressing embryos, and a Myc tag for detection by western blot. At 24 hr post injection, the GFP-positive embryos were dechorionated before incubation in biotin-supplemented media for a further 18 hr (*Figure 1A*). Total protein extracts from fish embryos were then subjected to SDS-PAGE and streptavidin immunoblotting (*Figure 1B*). TurboID injected embryos exhibited the strongest biotinylation of endogenous proteins among the three new biotin ligases with 500 µM biotin incubation. Of note, BASU and miniTurbo were expressed at a lower level than TurboID despite equal amount of RNA injection. While we did not explore the underlying reasons for this, greater instability of miniTurbo has been reported previously and other biotin ligases have shown poor expression in cell cultures (*May et al., 2020*; *Branon et al., 2018*). Therefore, we chose TurboID for all subsequent experiments. Note the two prominent bands consistently detected around 70 and 135 kDa in all samples likely represent endogenously biotinylated proteins (*Housley et al., 2014*; *Ahmed et al., 2014*).

To visualise TurboID-catalysed biotinylation in situ, TurboID-expressing embryos were stained with NeutrAvidin-DyLight 550 after biotin incubation (*Figure 1C*). The mosaic expression of TurboID-GFP in the muscle fibres, as well as expression in the yolk, corresponded with strong NeutrAvidin staining. The mRNA injections frequently gave rise to differing levels of expression between individual muscle cells within the same animal. Therefore, muscle fibres with little or no TurboID-GFP expression served as an internal negative control.

Biotin concentration and incubation time are two crucial factors that affect biotin ligase efficiency in cultured cells (*Roux et al., 2012*; *Kim et al., 2016*; *Branon et al., 2018*; *Ramanathan et al., 2018*). To achieve the most effective experimental conditions for TurboID application in zebrafish, we sought to optimise these parameters. From our initial experiments with BirA*, we knew that zebrafish embryos are able to tolerate a biotin concentration as high as 800 µM with no obvious morphological abnormalities (*Figure 1—figure supplement 2*). To determine the optimal biotin concentration for TurboID in zebrafish, TurboID-expressing embryos were incubated in embryo medium containing biotin concentration ranging from 0 to 750 µM for 18 hr, followed by lysis and

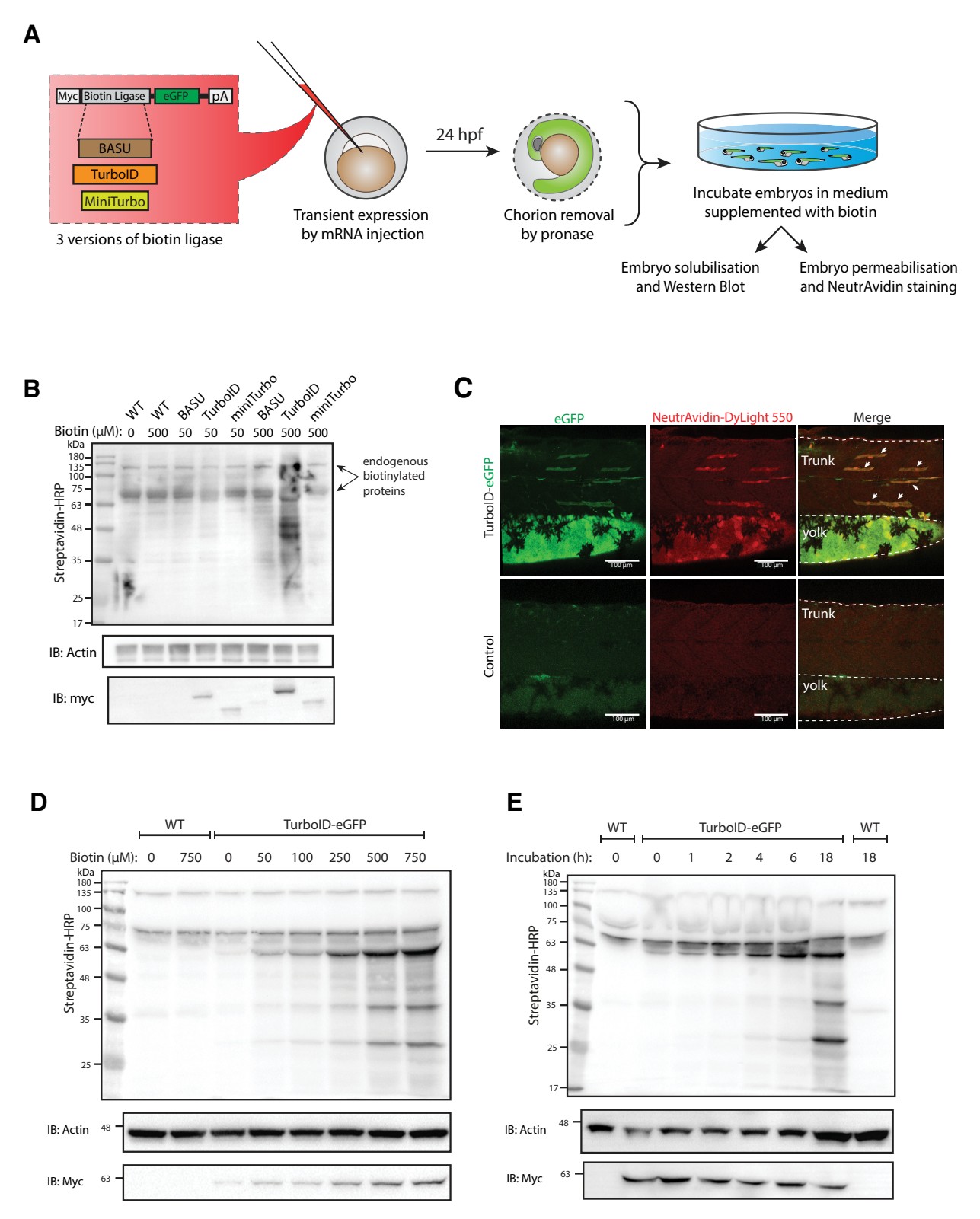

**Figure 1.** Testing and optimising biotin ligases: BASU, TurboID, and miniTurbo. (**A**) A schematic overview of the workflow. The BASU/TurboID/MiniTurbo was transiently expressed in zebrafish embryos by RNA injection. Chorion-removed fish embryos with green fluorescence were selected for incubation in biotin supplemented embryo media for 18 hr. After biotin incubation, embryos were analysed by western blotting and immunofluorescence. (**B**) The streptavidin-HRP blot showing biotinylated proteins in two dpf zebrafish embryos expressing eGFP-tagged BASU,

*Figure 1 continued on next page*

*Figure 1 continued*

TurboID, and miniTurbo. Fish embryos were incubated in biotin concentrations of 50 or 500 µM biotin for 18 hr before embryo solubilisation and Western blot analysis. Actin immunoblot (IB:Actin) serves as a loading control; the anti-Myc immunoblot (IB:Myc) reflects the protein level of each biotin ligases; each sample is a pool of 30 embryos. (C) Representative images of NeutrAvidin staining of biotinylated proteins in 2 dpf zebrafish embryo transiently expressing TurboID-eGFP. Fish muscle and yolk were outlined with dashed lines. White arrows indicate muscle fibres expressing TurboID-eGFP. n = 6. Scale bar denotes 100 µm (D and E) Dependency of TurboID activity on biotin concentration and incubation time. Zebrafish embryos transiently expressing TurboID-eGFP were incubated with embryo media containing 0 to 750 µM biotin for 18 hr (D) or incubated with 500 µM biotin for 0 to 18 hr (E) before protein extraction and immunoblot analysis with streptavidin-HRP, anti-Actin and anti-Myc antibodies; each sample is a pool of 30 embryos. For immunoblots showing the biotinylation of BioID and BioID2 in zebrafish embryos see *Figure 1—figure supplement 1*. For biotin tolerance of zebrafish embryos see *Figure 1—figure supplement 2*. For original western blot images see *Figure 1—source data 1*.

The online version of this article includes the following source data and figure supplement(s) for figure 1:

**Source data 1.** Raw images of blots.
**Figure supplement 1.** Testing BioID and BioID2 in transgenic zebrafish.
**Figure supplement 2.** Determining biotin toxicity and tolerance in zebrafish embryos.

streptavidin immunoblotting. Weak labelling could be seen with 50 µM biotin, increasing through 250 µM, with the strongest labelling at concentration of 500 and 750 µM (*Figure 1D*). Unlike its application in cultured cells and yeast (*Branon et al., 2018*), TurboID did not produce detectable exogenous biotinylation without the addition of biotin (*Figure 1D*). This provides the opportunity for temporal resolution by addition of exogenous biotin at specific developmental stages. Unexpectedly, the anti-Myc immunoblot showed that a higher biotin concentration resulted in more TurboID in the total protein extracts (*Figure 1D*). Concomitantly, the addition of exogenous biotin did not change the level of endogenous biotinylated proteins in the WT embryos (*Figure 1D*).

In mammalian cell culture, a 10-min biotin incubation with TurboID is sufficient to visualise biotinylated proteins by immunoblotting and to perform analysis of different organellar proteomes (*Branon et al., 2018*). However, we did not observe rapid biotinylation in zebrafish within the first 2 hr of biotin incubation (*Figure 1E*). TurboID-induced biotin labelling was only weakly detected after 4–6 hr incubation and adequate biotinylation was only detected after overnight incubation (18 hr).

## In vivo proximity biotinylation targeted to a specific subcellular region or a protein of interest

Next, we tested the spatial resolution of TurboID-catalysed biotinylation in zebrafish when TurboID was targeted to a specific subcellular region and to a POI. We tagged TurboID with a nuclear localisation signal (NLS), a plasma membrane localisation motif (CaaX), the transmembrane protein CD44b and the muscle T-tubule enriched membrane protein Cavin4b (*Figure 2A*). After biotin treatment, the TurboID fusion proteins produced a biotinylation pattern corresponding to the appropriate subcellular location of targeting sequences/proteins in zebrafish embryos (*Figure 2B*). The spatial resolution of the biotin labelling was remarkable as even the T-tubule structure, which is difficult to resolve in fixed embryos, was clearly visible by NeutrAvidin staining in the embryos expressing Cavin4b-TurboID. Furthermore, the biotinylated protein derived from each TurboID construct gave rise to a unique barcode of protein bands on the streptavidin blot, indicative of proteins specific to each corresponding subcellular compartment (*Figure 2C*). These results demonstrated that TurboID was able to specifically label a selective subpopulation of endogenous proteins when targeted to a specific subcellular region or protein in zebrafish embryos. Moreover, the TurboID-biotinylated proteins were recoverable from crude fish lysates by affinity purification with streptavidin-conjugated beads (*Figure 2D*), ready for downstream applications such as identification by mass spectrometry.

Overall, TurboID showed robust biotin labelling with high spatial resolution in zebrafish embryos. These properties rendered it suitable for pursuing in vivo proteomic analyses.

## Conditionally stabilised GFP-binding protein (dGBP) is able to target GFP-tagged proteins in zebrafish

Although we were able to achieve proximity-dependent biotin labelling in zebrafish embryos transiently expressing TurboID by mRNA injection, this method requires the direct injection of a large number of newly fertilised embryos in order to obtain sufficient protein for subsequent mass

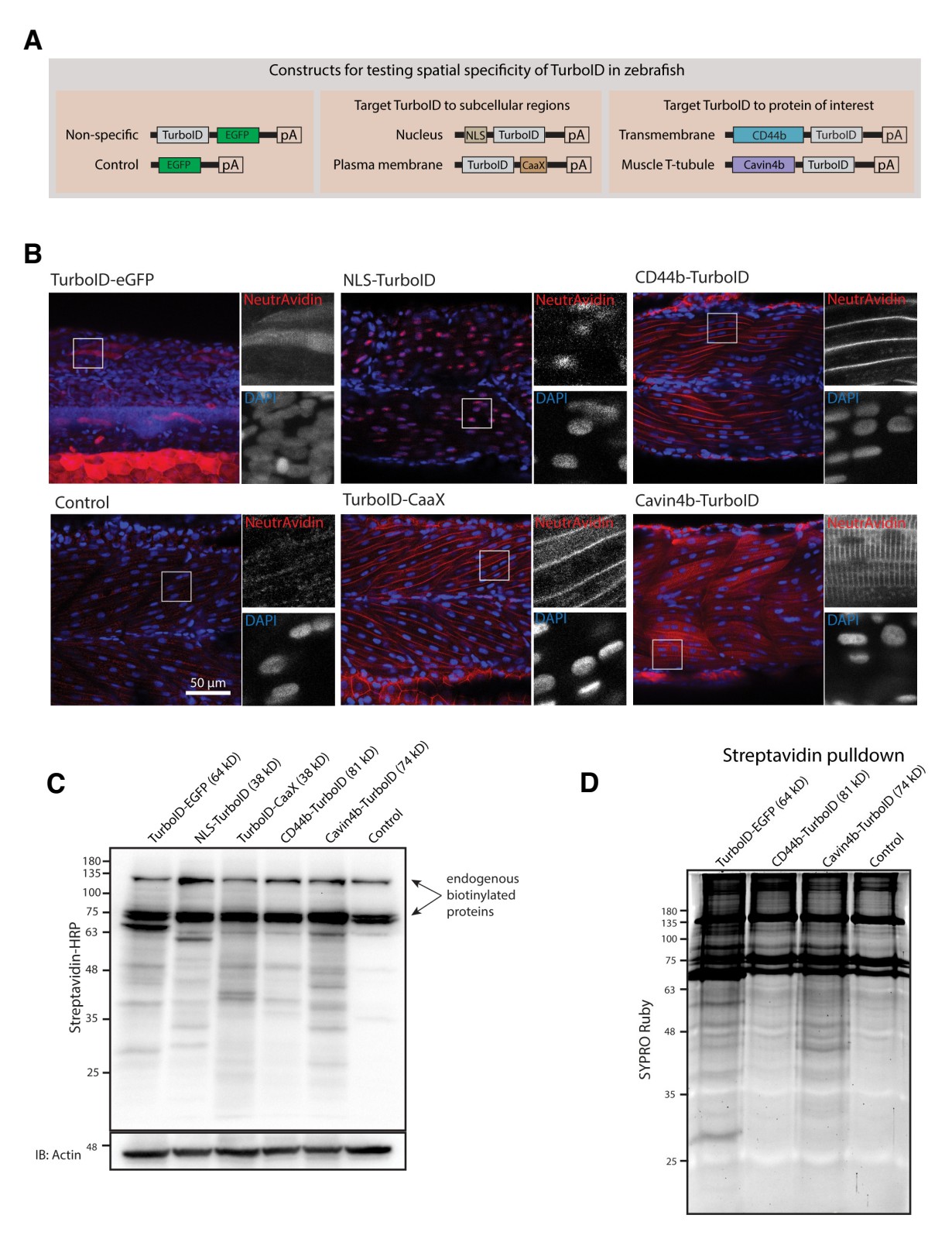

**Figure 2.** Spatial resolution of TurboID-catalysed biotinylation in zebrafish embryos. (**A**) Schematic representation of eGFP-, NLS-, CaaX-, CD44b-, and Cavin4b-tagged TurboID constructs for mRNA injection in zebrafish embryos. TurboID-eGFP was used as a positive control. (**B**) Representative images showing the distribution of biotinylated proteins in two dpf zebrafish embryos transiently expressing different TurboID constructs. Negative control fish were injected with eGFP only. Fish embryos were fixed and permeabilised before NeutrAvidin-DyLight staining for biotin and DAPI staining to indicate

*Figure 2 continued on next page*

*Figure 2 continued*

nuclei. Regions within the white box were magnified and shown in the gray scale for NeutrAvidin and DAPI staining in the right panel; n = 3. Scale bar represents 50 µm. (C) Streptavidin-HRP blots showing the 'protein barcode' produced by biotinylated proteins in fish embryo transiently expressing different TurboID constructs. Actin immunoblot served as a loading control. Each sample is a pool of 30 embryos. (D) SYPRO Ruby protein stain showing proteins isolated by streptavidin-pulldown. Approximately three hundred embryos transiently expressing each TurboID constructs were subjected to streptavidin-pulldown after biotin incubation and embryo lysis. For original western blot/gel images see *Figure 2—source data 1*. The online version of this article includes the following source data for figure 2:

**Source data 1.** Raw images of blots.

spectrometry sequencing. It is a labour-intensive exercise when potentially analysing multiple POIs, and new genetic constructs must be generated for each POI. In addition, the protein expressed from mRNA injected at the one-cell stage becomes progressively depleted and is present only in trace amounts beyond 3 days post fertilisation. As such, this methodology is limited to early stage embryos. To circumvent these issues, we envisaged a modular system that would utilise the many existing stable zebrafish lines which express GFP-tagged proteins. Previously, we demonstrated that a GFP-binding peptide (GBP; a 14 kDa nanobody) is able to target a peroxidase (APEX2) to GFP-tagged POIs in both cell culture and zebrafish systems (*Ariotti et al., 2015b*), and can be used for ultrastructural localisation. Based on these findings, we reasoned that genetically fusing TurboID with GBP would target the TurboID-GBP fusion protein to GFP-labelled POIs and/or subcellular compartments in zebrafish, enabling GFP-directed proximity biotinylation in vivo. Furthermore, generation of a stable zebrafish line expressing TurboID-GBP would allow delivery of the transgene by a simple genetic cross, circumventing the need for microinjection and enabling continued expression beyond the embryonic stages.

As proof-of-principle, we fused a red fluorescent protein (mRuby2) with the GBP nanobody and transiently expressed it in a transgenic fish line already expressing Cavin1a-Clover. Clover is a GFP derivative recognised by GBP (*Shaner et al., 2013*), and Cavin1a is an ortholog of caveolae-associated protein one in zebrafish (*Lo et al., 2015*). When expressed at low levels, mRuby2-GBP showed clear colocalisation with Cavin1a-Clover at the plasma membrane in the mRuby2-positive muscle cells (*Figure 3A*). However, when mRuby-GBP was expressed at higher levels, red fluorescence was observed in the cytoplasm in addition to the plasma membrane, likely due to the saturation of binding between GBP and GFP. This observation raised concerns about the potential of non-specific labelling from unbound TurboID-GBP under these conditions. As a solution, we substituted the GBP with a conditionally stabilised GFP-nanobody (destabilised GBP or 'dGBP') that is rapidly degraded unless the GFP-binding site is occupied (*Tang et al., 2016*; *Ariotti et al., 2018a*). Using this approach, we observed tight association of mRuby2-dGBP and Cavin1a-Clover in all muscle cells regardless of expression level (*Figure 3B*). We reasoned that a system utilising the conditionally stabilised nanobody would be less likely to result in non-specific biotin labelling within target cells in vivo. Furthermore, use of the conditionally stabilised GBP gives potential for modularity, since tissue or cell type specific biotinylation will only occur in cells expressing both GFP-POI and TurboID-dGBP fusion proteins.

## Development of BLITZ; biotin labelling in tagged zebrafish

We next generated a number of fish lines expressing TurboID-dGBP under the ubiquitous beta actin 2 (*actb2*) promoter (*Casadei et al., 2011*). To facilitate selection of appropriate transgenic integrations, we added a cytoplasmic red fluorescent protein, mKate2, as a visible reporter upstream of TurboID-dGBP linked by a P2A sequence (*Donnelly et al., 2001*; *Kim et al., 2011*). The P2A sequence is a short ribosome-skipping sequence which separates the upstream mKate2 from downstream TurboID-dGBP, reducing the potential interference from the fluorescent protein. The expression of transgene is stable in our zebrafish lines and demonstrates Mendelian inheritance over four generations, indicating a stable single transgenic integration.

We first tested whether these zebrafish lines were able to catalyse specific biotinylation in tissues expressing GFP. The TurboID-dGBP fish were outcrossed with transgenic lines expressing cytoplasmic GFP in the vasculature (kdrl:EGFP) (*Beis et al., 2005*) and the motor neurons (MotoN:EGFP) (*Punnamoottil et al., 2015*; *Figure 4A*). Biotinylated proteins were examined in three dpf embryos after overnight biotin incubation. In embryos co-expressing ubiquitous TurboID-dGBP and tissue-

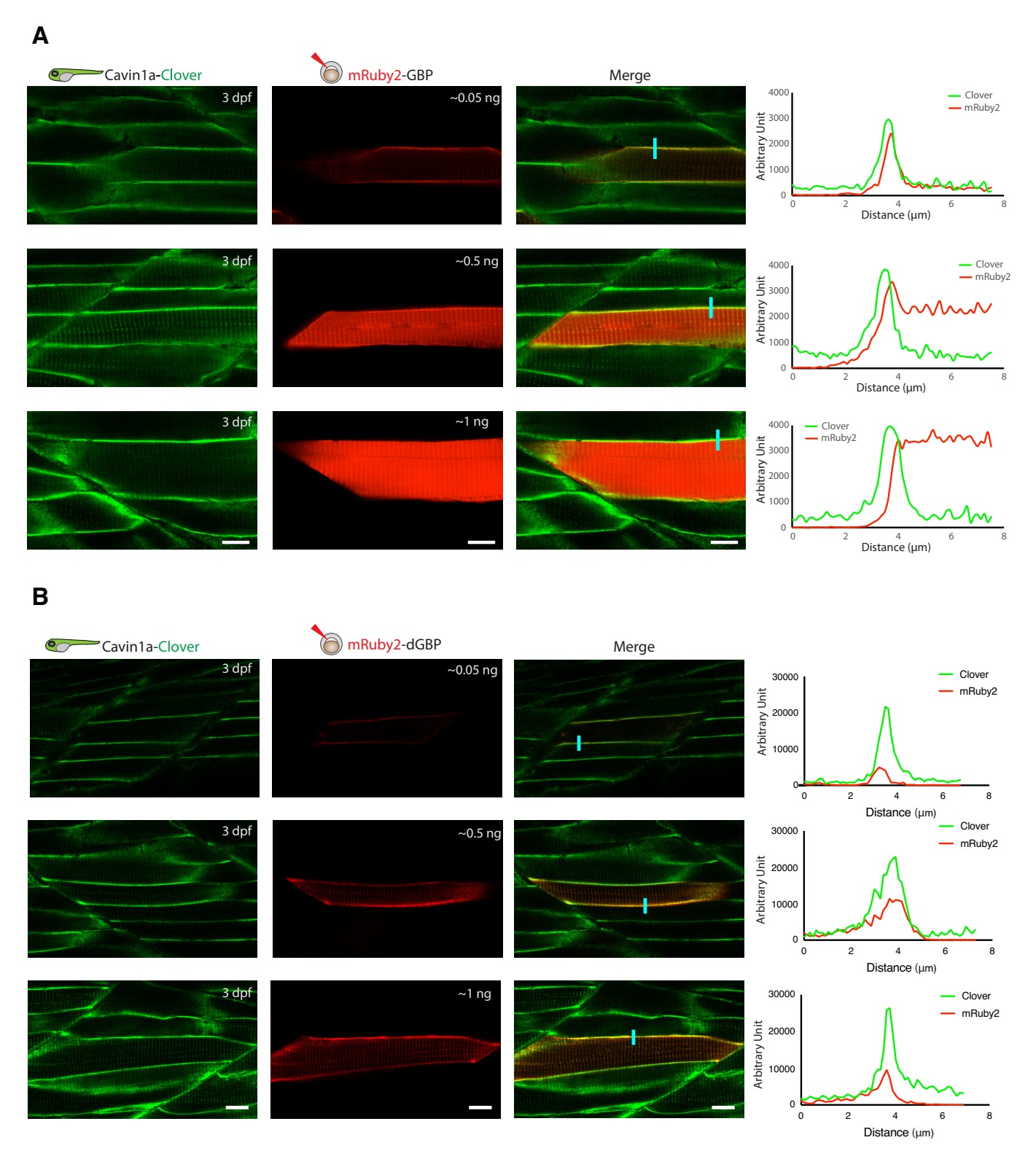

**Figure 3.** In vivo binding of GFP-nanobody, GBP and dGBP, in stable transgenic zebrafish embryos. (**A and B**) Representative images showing the colocalisation between Cavin1a-Clover and mRuby2-GBP/dGBP in live zebrafish embryos. Cavin1a-Clover zebrafish embryos transiently expressing mRuby2-tagged GBP (**A**) or dGBP (**B**). Injected embryos were imaged at three dpf. The approximate amount of injected RNA was indicated in the

*Figure 3 continued on next page*

*Figure 3 continued*

mRuby2 images. Line scan (indicated by the blue line) shows the fluorescent intensity of Clover and mRuby2 across the sarcolemma of mRuby2-positive muscle cells. Scale bar denotes 10 μm in both (**A**) and (**B**).

specific GFP, TurboID-catalysed biotinylation was detected in the intersegmental vessels and the spinal cord motor neurons in the kdrl:EGFP and MotoN:EGFP lines, respectively (*Figure 4A*). These results demonstrate that our TurboID-dGBP system can produce biotinylation with tissue specificity.

To test the biotinylation on a subcellular level, the TurboID-dGBP fish were outcrossed with Cavin1a-Clover and Cavin4a-Clover transgenic fish lines expressing Cavin1a-Clover and Cavin4a-Clover under the control of the muscle specific actin promoter, *actc1b*. Cavin1a and Cavin4a are orthologues of human CAVIN1 and CAVIN4, which are caveola-associated proteins involved in caveolar formation. With the same procedures, we observed clear colocalisation between biotinylated proteins and Clover-tagged cavins in muscle fibres, at the sarcolemma and T-tubules, suggesting our TurboID-dGBP system can produce proximity-dependent biotinylation with subcellular resolution. Without biotin treatment or without the expression of GFP, there was no detectable biotinylation effected by TurboID. Notably, the specificity of GFP-directed biotinylation was not compromised in fish lines expressing higher levels of TurboID-dGBP (*Figure 4—figure supplement 1*).

We next visualised the proteins biotinylated by TurboID-dGBP on streptavidin blots (*Figure 4B*). The two prominent bands representing endogenously biotinylated proteins were again observed in embryos carrying both TurboID-dGBP and Cavin1a-clover; omitting the biotin supplement resulted in no exogenous biotinylation. Intriguingly, in the absence of Cavin1a-Clover, a weak biotinylation was still observed in the embryos carrying only the TurboID-dGBP transgene, despite the level of TurboID-dGBP being undetectable on anti-Myc immunoblot. This background labelling is likely caused by TurboID-dGBP en route to proteasomal degradation. Subsequent MS analysis revealed these background proteins are mainly endogenous biotinylated proteins, nuclear proteins, cytoskeletal proteins and yolk proteins (*Supplementary file 3*). Using streptavidin affinity pulldown, biotinylated proteins were isolated from total fish lysates and endogenous Cavin4b, a known Cavin1 interactor (*Bastiani et al., 2009*), was detected in the streptavidin pulldown in addition to Cavin1a-GFP and TurboID-dGBP (*Figure 4C*). Note that a trace of TurboID-dGBP was detected in the streptavidin pulldown in the absence of GFP target with long exposure, which accounts for the weak background biotinylation in embryos expressing only TurboID-dGBP.

## A comprehensive cavin-associated proteome in skeletal muscle generated by TurboID-dGBP

Finally, we employed our TurboID-dGBP system to map the proteomes associated with Cavin1 and 4 in zebrafish skeletal muscle. We crossed the TurboID-dGBP fish with fish lines stably expressing Cavin1a-Clover, Cavin4a-Clover, and Cavin4b-Clover in muscle. TurboID-dGBP and cavin-Clover co-expressing embryos were selected at 2 dpf for subsequent biotin labelling and the biotinylated proteins were analysed by liquid chromatography coupled to tandem mass spectrometry (nanoHPLC/MS MS/MS). The sibling embryos inheriting only the TurboID transgene were used as a background control (experimental regimen illustrated *Figure 5A*). After subtracting background proteins and common containments, 26, 22, and 25 proteins were identified in the Cavin1a, Cavin4a, and Cavin4b samples, respectively (*Figure 5B and C*, *Supplementary file 1* – Tables 1-3). Among the proteins identified, the majority of proteins are associated with the plasma membrane, consistent with the membrane localisation of cavins (*Figure 5D*, *Supplementary file 1* - Table 4). Endogenous cavins were consistently detected in all samples, suggesting that the GFP-tag, as well as the binding of TurboID-dGBP, did not interfere with the oligomerisation of the cavins. Dystrophin (DMD), a protein associated with caveolae was identified uniquely in the Cavin1a sample (*Song et al., 1996*; *Doyle et al., 2000*). An ortholog of Pacsin3 (zgc:91999), a caveola-associated protein required for muscle caveolar formation (*Seemann et al., 2017*), was also detected uniquely in the Cavin4b sample. These known interactors were undetectable in all control samples, demonstrating the high accuracy of the BLITZ system. Note that, Caveolin1 (Uniprot accession: Q6YLH8) and Caveolin (isoform unassigned; Uniprot accession: A1L1S3) were detected in the Cavin1a sample based on two high confidence peptides (HLNDDVVK and VWVYSGIGFESAR) that were not present in the controls

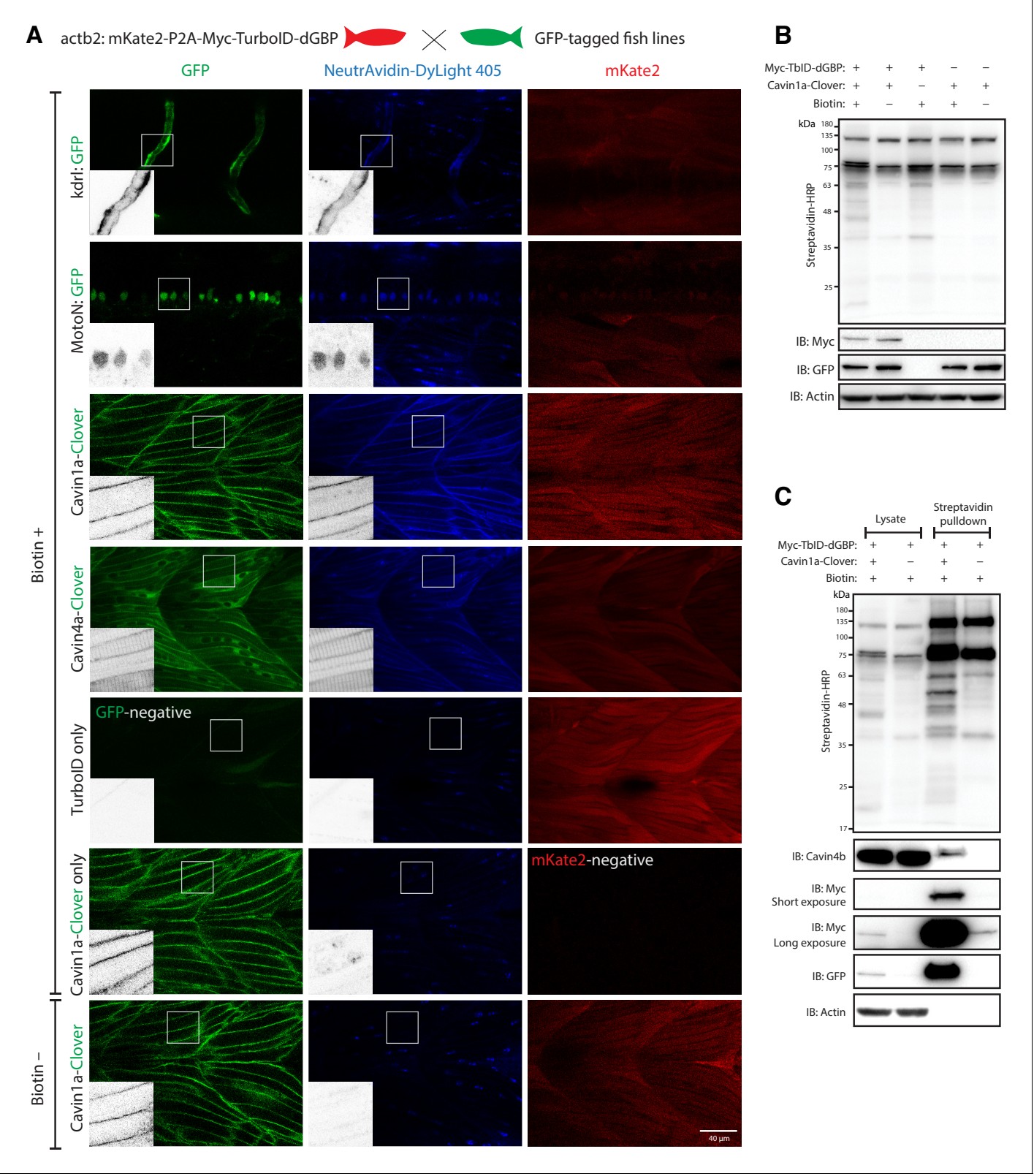

**Figure 4.** GFP-directed in vivo biotin labelling. (**A**) Representative images of TurboID-dGBP catalysing GFP-dependent biotinylation in transgenic zebrafish embryos at 3 dpf. The TurboID-dGBP line was crossed with different GFP-tagged zebrafish lines: Cavin1a-Clover (plasma membrane), Cavin4a-Clover (sarcolemma and T-tubules), kdrl:eGFP (vasculature), and MotoN:eGFP (motor neurons). After biotin incubation, embryos were fixed, permeabilised, and stained with NeutrAvidin to visualise the biotinylated protein. mKate2 is a fluorescent indicator for expression of TurboID-dGBP.
*Figure 4 continued on next page*

*Figure 4 continued*

Controls were carried out by using siblings from the same clutch without GFP expression (TurboID only) and siblings without TurboID expression (Cavin1a-Clover only), as well as omitting biotin incubation. The scale bar denotes 40 µm; n = 3. (B) Western blot analysis showing the biotinylated proteins in 3 dpf zebrafish embryos from TurboID-dGBP outcrossing with Cavin1a-Clover line. Each sample is a pool of 30 embryos. (C) Western blot analysis of fish lysates and streptavidin pulldown with embryos from TurboID-dGBP line outcrossing with Cavin1a-Clover line. Each pulldown sample is a pool of 200 embryos. For confocal images comparing the biotin labelling specificity in zebrafish embryos with different expression level of TurboID-dGBP see *Figure 4—figure supplement 1*. For table summarising proteins identified in control embryos expressing only TurboID-dGBP, see *Supplementary file 3*. For original western blot images see *Figure 4—source data 1*.

The online version of this article includes the following source data and figure supplement(s) for figure 4:

**Source data 1.** Raw images of blots.

**Figure supplement 1.** The specificity of TurboID-dGBP biotin labelling is independent of its expression level in zebrafish embryos.

(*Supplementary file 2*) but were not classified as significant hits based on the 1% global FDR analysis. This may reflect the poor accessibility of caveolins, as a large proportion of caveolins are buried in the plasma membrane (*Ariotti et al., 2015a*). Individual hits and general properties of putative interacting proteins are further discussed below.

## Discussion

### Advantages of BLITZ in proteomic mapping

In this study, we have developed BLITZ (Biotin Labelling In Tagged Zebrafish): a modular system for in vivo proteomic mapping (*Figure 6*). This system utilises the advantages of BioID at capturing weak or transient interactions in living cells, but extends its application to an in vivo setting, enabling interactome investigation at specific developmental stages under physiological conditions, and potentially in disease models. The system also has several advantages over conventional BioID methods for studying the proteome and interactome.

Firstly, BLITZ does not require extensive molecular biology steps to produce numerous expression constructs, or laborious embryonic manipulation. It instead relies on simple crossing of a TurboID-dGBP line with an existing GFP-tagged fish line of choice; a plethora of such lines currently exist in stock centres globally and, with the advent of nuclease directed genome editing, this number is rapidly increasing. Secondly, BLITZ enables cell- and tissue-specific proteomic studies, since the stability of TurboID-dGBP is dependent on its binding to GFP targets. Non-specific biotin labelling in tissues that do not express GFP-tagged constructs is avoided. Thirdly, our TurboID-dGBP system has the potential to be used with knock-in fish lines carrying a GFP fusion protein at the endogenous locus of a POI. This will enable the application of BioID to study proteomic associations with endogenous proteins, which, to our knowledge, has not been achieved by conventional BioID methods. Finally, the modularity of the BLITZ system could be advantageous for use in established tissue culture systems using existing GFP expression vectors or/and knock-in cell lines, as well as extended to other organisms.

The use of the BLITZ system also comes with some caveats. Unlike the traditional BioID approach using a direct fusion of the biotin ligase with the bait protein, our system targets TurboID to the POI through the binding of dGBP nanobody to GFP. In this case, the indirect binding increases the physical distance between biotin ligase and the POI, which could potentially enlarge the effective labelling radius and include more non-interacting neighbouring proteins. However, we have previously shown that the use of a GFP-directed nanobody to target a genetically encoded peroxidase (APEX2) for protein localisation does not appear to compromise the fidelity of labelling: APEX2 staining was rarely observed beyond 25 nm from the site of POI (*Ariotti et al., 2015a*). It is also possible that the binding of the biotin ligase-nanobody with the GFP-tagged POI could perturb the localisation of the POI, either by masking interacting surfaces or simply due to the larger size of a complex. For this reason, we routinely examine the distribution of the GFP-tagged POI both with and without biotin ligase-dGBP expression as well as the distribution of biotinylated proteins using fluorescent neutravidin staining. Since our in vivo system is based on the simple crossing of heterozygous transgenic lines, every new clutch contains offspring with every possible combination of alleles, and the appropriate internal controls can be sorted by fluorescence. As BLITZ uses biotin as a label, the method

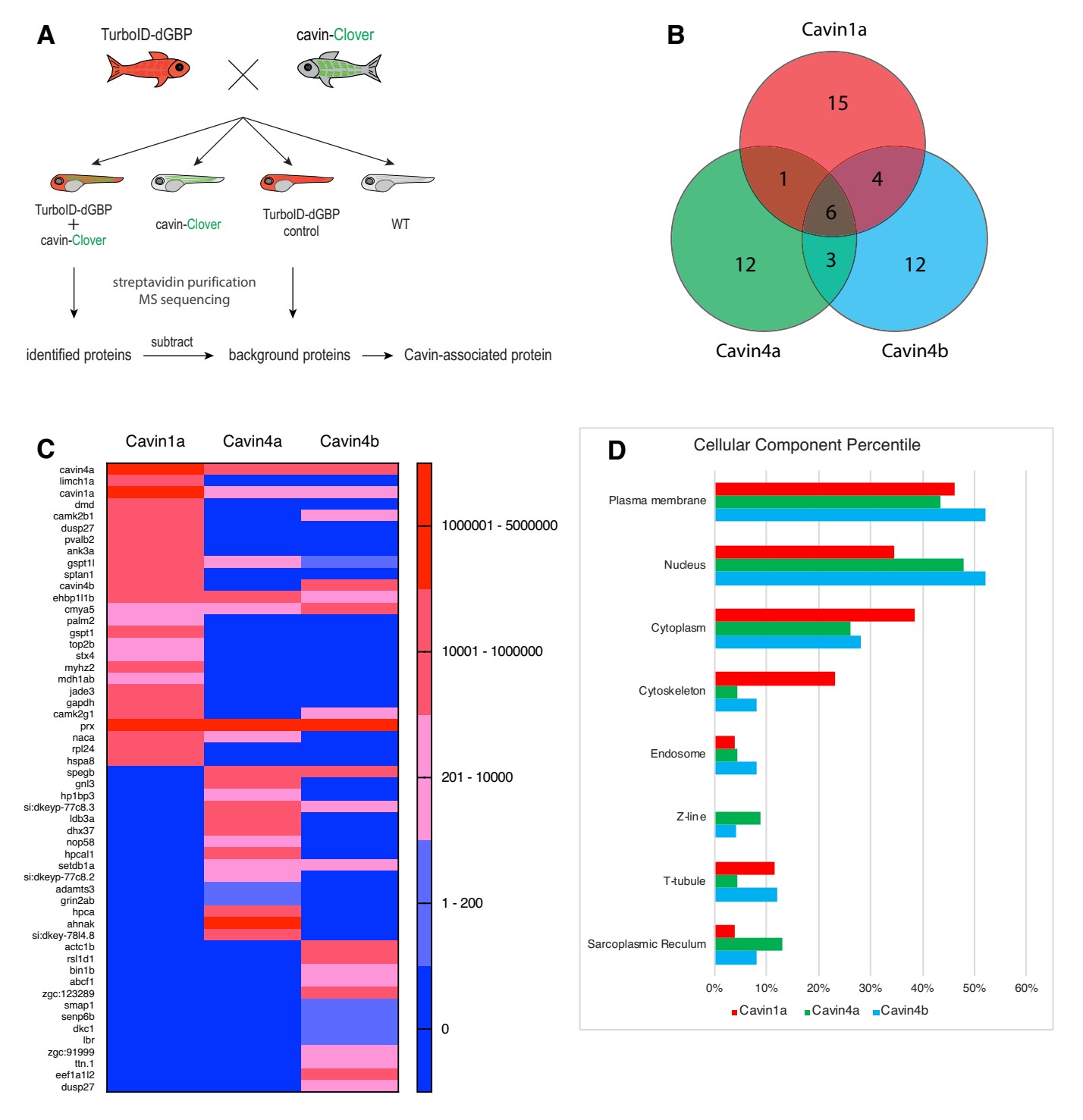

**Figure 5.** Proteomes identified by BLITZ system in Clover-tagged cavin zebrafish. (**A**) A schematic overview of applying TurboID-dGBP fish to identify cavin-associated proteins. The TurboID-dGBP zebrafish was crossed with Clover-tagged cavin fish lines. The embryos carrying both transgenes were selected for subsequent biotin incubation and biotin affinity purification coupled MS sequencing. Identified proteins were refined by subtracting proteins identified in control embryos expressing only TurboID-dGBP. (**B**) Venn diagram showing the overlap of identified proteins in Cavin1a, Cavin4a, and Cavin4b samples. (**C**) Heatmap showing relative abundance of identified proteins based on normalised MS2Count in Cavin1a, Cavin4a, and Cavin4b proteomes. (**D**) Bar graph showing the distribution of proteins at subcellular level. The cellular component information was curated from Uniport database and literature. For table summarising all identified and enriched proteins, see *Supplementary file 1* – Tables 1-3. For table annotating all identified and enriched protein with subcellular localisation and functions, see *Supplementary file 1* – Table 4. For table showing all

*Figure 5 continued on next page*

*Figure 5 continued*

identified peptides in Cavin1a sample and sibling control sample see *Supplementary file 2* – Tables 1-2. For protein identification report generated by ProteinPilot, see *Figure 5—source datas 1–6*.

The online version of this article includes the following source data for figure 5:

**Source data 1.** Protein identification report for Cavin1a sample generated by ProteinPilot.
**Source data 2.** Protein identification report for Cavin1a control sample generated by ProteinPilot.
**Source data 3.** Protein identification report for Cavin4a sample generated by ProteinPilot.
**Source data 4.** Protein identification report for Cavin4a control sample generated by ProteinPilot.
**Source data 5.** Protein identification report for Cavin4b sample generated by ProteinPilot.
**Source data 6.** Protein identification report for Cavin4b control sample generated by ProteinPilot.

(like most BioID methods) is problematic for the identification of interactors that are endogenously biotinylated, such as carboxylases. In addition, as all BioID methods will label proteins within a small number of nanometres, non-interacting proteins could be detected simply due to close proximity. Thus, subsequent validation using other independent approaches such as biomolecular fluorescence complementation and affinity pulldown is essential to distinguish bona fide interactors from non-interacting neighbouring proteins.

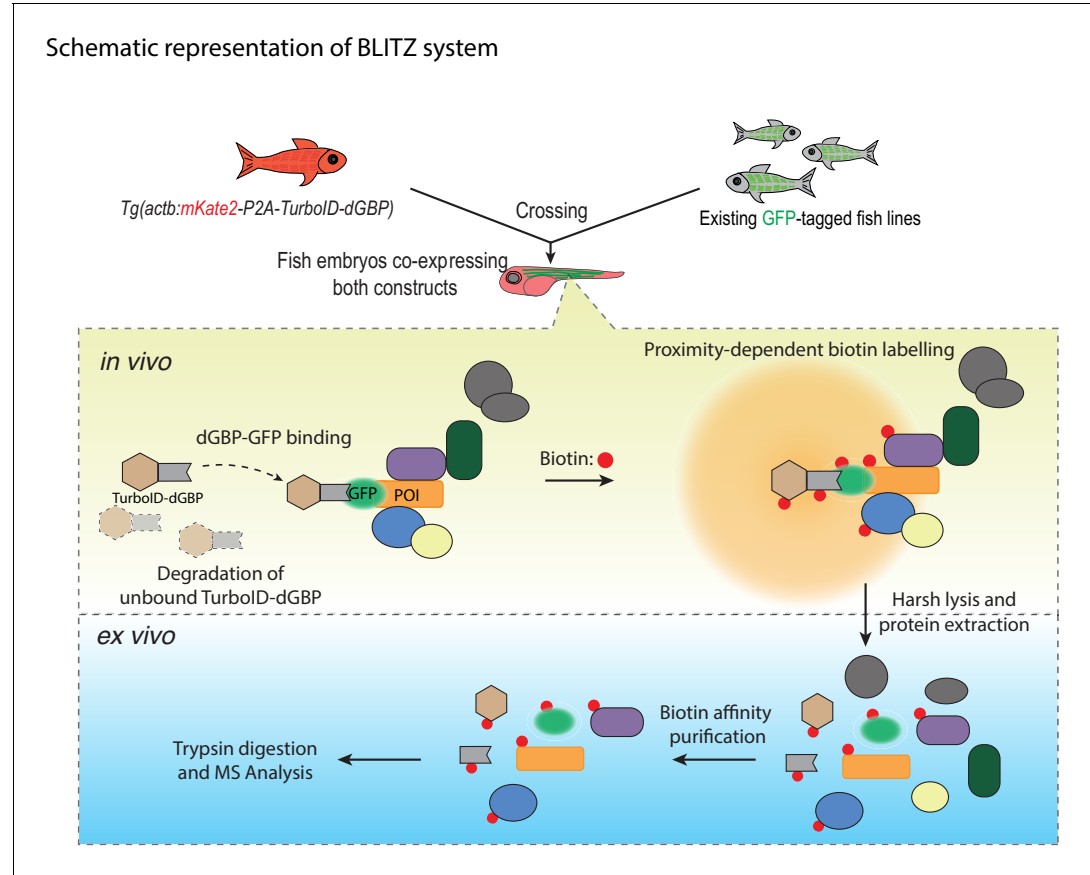

**Figure 6.** A schematic overview of the BLITZ system. The TurboID-dGBP lines can be crossed with existing GFP-tagged lines. In the embryos carrying both transgenes, the binding between dGBP and GFP stabilise TurboID-dGBP, which leads to proximity biotinylation around the GFP-tagged POIs. The unbound TurboID-dGBP will be rapidly degraded by the ubiquitin proteasome system, which minimises non-specific labelling when dGBP-GFP binding saturates, as well as achieving tissue specificity by averting labelling in cells/tissues that do not express GFP. The biotin-labelled proteins can be isolated by biotin affinity purification and identified by MS analysis.

## Application of BLITZ to the identification of cavin-association networks in muscle

Cavin family proteins are key components of the caveolar coat complex associated with the inner leaflet of the plasma membrane. Cavin1 is present in all tissues and is essential for caveolar formation and function. Cavin2, 3, and 4 show more restricted tissue distributions with Cavin4 being specific to skeletal and cardiac muscle (reviewed in *Parton et al., 2018*). In the zebrafish, Cavin1 and 4 are each duplicated such that four loci exist; Cavin1a/b and Cavin4a/b. Cavin1a and b show spatially distinct expression patterns with Cavin1b being largely restricted to the developing notochord whereas Cavin1a, 4a, and 4b are all highly expressed in skeletal muscle (*Hill et al., 2008*; *Lo et al., 2015*; *Housley et al., 2016*). In this study we used the BLITZ system to identify putative interactors for all three skeletal muscle cavins, and identified sets of putative interactors both unique and common to all three proteins.

The majority of proteins identified for all cavin proteomes were muscle-enriched factors and plasma membrane proteins. We also saw a specific enrichment of known caveola-associated proteins consistent with initial expectations. Interestingly, the cavin proteomes also contained a disproportionate number of nuclear proteins, such as Gnl3, Naca, and Lbr. In cultured cells, cavins have been shown to be released from the plasma membrane in response to external stimuli (such as mechanical stress) and are able to bind intracellular targets in variety of subcellular locations to regulate processes such as ribosomal RNA transcription and apoptosis (*Liu and Pilch, 2016*; *McMahon et al., 2019*). In addition, in the absence of Cavin1, in knockout mouse muscle, Cavin4 has been shown to localise predominantly to the nucleus rather than the sarcolemma (*Lo et al., 2015*).

What processes might Cavin1 and Cavin4 be regulating? We know that loss of Cavin1 causes lipodystrophy and muscular dystrophy in humans. Patient and animal muscle shows hypertrophied muscle fibres (*Hayashi et al., 2009*; *Rajab et al., 2010*; *Ding et al., 2017*). Cavin4 mutations have been described in dilated cardiomyopathy patients and there is evidence that Cavin4 recruits ERK in cardiomyocytes (*Rodriguez et al., 2011*; *Ogata et al., 2014*). Thus, there is supporting data for the positive regulation of hypertrophy in skeletal muscle fibres by Cavin1, and in cardiomyocytes by Cavin4. The cavin proteome showed an enrichment of protein kinases, such as calcium/calmodulin-dependent protein kinase II (CaMKII). CaMKII regulates $Ca^{2+}$ signalling and plays an important role in the development of cardiac hypertrophy through the ERK signalling pathway (*Illario et al., 2003*; *Cipolletta et al., 2010*; *Cipolletta et al., 2015*). The activation of CaMKII can be induced by exercise in skeletal muscle, with the activation level proportional to the intensity of exercise (*Rose et al., 2006*).

In this study, BLITZ revealed several putative cavin interactors that have also been shown to be involved in cardiomyopathies and/or skeletal myopathies, including the membrane protein Dystrophin (*Deconinck and Dan, 2007*), the triad-associated proteins Bin1 (*Nicot et al., 2007*), Cypher/ZASP (*Selcen and Engel, 2005*), and SPEG (*Agrawal et al., 2014*). Genetic ablation of zebrafish Cavin4b causes aberrant T-tubules in skeletal muscle (*Housley et al., 2016*). It is possible that Cavin4 may be involved in T-tubule formation through interaction with triad associated proteins, such as Bin1.

Overall, our BLITZ system enables the in vivo identification of protein interactors in a cell- and tissue-specific manner, with high precision. We demonstrated the applicability of this approach in diverse cell types including neurons and vascular endothelial cells and applied the BLITZ system to identify factors associated with cavin family proteins in differentiated skeletal muscle. BLITZ provides a versatile and valuable tool for proteomic discovery using the zebrafish model, but also has the potential for application in other in vivo contexts that to date have been challenging or intractable.

## Materials and methods

**Key resources table**

| Reagent type (species) or resource | Designation | Source or reference | Identifiers | Additional information |
|---|---|---|---|---|
| Gene (*E. coli-* modified) | BirA* | *Roux et al., 2012*; DOI: 10.1083/jcb.201112098 | | R118G mutant of WT BirA |

*Continued on next page*

*Continued*

| Reagent type (species) or resource | Designation | Source or reference | Identifiers | Additional information |
|---|---|---|---|---|
| Gene (*Bacillus subtilis* - modified) | BASU | *Ramanathan et al., 2018*; DOI: 10.1038/NMETH.4601 | | R124G, E323S, G325R mutation and N-terminus deletion of WT biotin ligase from *B. subtilis* |
| Gene (*Aquifex aeolicus* - modified) | BioID2 | *Kim et al., 2016*; DOI: 10.1091/mbc.E15-12-0844 | | R40G mutation of WT biotin ligase from A. aeolicus |
| Gene (*E. coli*- modified) | miniTurbo | *Branon et al., 2018*; DOI: 10.1038/nbt.4201 | | 13 point mutations and N-terminal deletion of WT BriA |
| Gene (*E. coli*- modified) | TurboID | *Branon et al., 2018*; DOI: 10.1038/nbt.4201 | | 15 point mutations of WT BirA |
| Strain, strain background (*Danio rerio*) | TAB | University of Queensland (UQ) Biological Resources Aquatics | | Wild-type (TAB), an AB/TU line generated in UQBR Aquatics (UQ Biological Resources) |
| Strain, strain background (*Danio rerio*) | TurboID-dGBP | Generated in this paper | | Tg(actb2:mKate2 -P2A-TurboID-dGBP) |
| Strain, strain background (*Danio rerio*) | Cavin1a-Clover | Generated in this paper | | Tg(actc1b:Cavin1a-Clover) |
| Strain, strain background (*Danio rerio*) | Cavin4a-Clover | Generated in this paper | | Tg(actc1b:Cavin4a-Clover) |
| Strain, strain background (*Danio rerio*) | Cavin4b-Clover | Generated in this paper | | Tg(actc1b:Cavin4b-Clover) |
| Strain, strain background (*Danio rerio*) | Kdrl:GFP | (*Beis et al., 2005*); DOI: 10.1242/dev.01970 | | Tg(kdrl:EGFP) |
| Strain, strain background (*Danio rerio*) | MotoN:GFP | (*Punnamoottil et al., 2015*); DOI: 10.1002/dvg.22852 | | Tg(miR218:EGFP) |
| Genetic reagent (*Danio rerio*) | actb2:mKate2-P2A-TurboID-dGBP | Generated in this paper | Addgene: 163857 | Construct for generating stable transgenic fish line; see Materials and methods for line generation |
| Genetic reagent (*Danio rerio*) | actc1b:Cavin1a-Clover | Generated in this paper | Addgene: 163852 | Construct for generating stable transgenic fish line; see Materials and methods for line generation |
| Genetic reagent (*Danio rerio*) | actc1b:Cavin4a-Clover | Generated in this paper | Addgene: 163853 | Construct for generating stable transgenic fish line; see Materials and methods for line generation |
| Genetic reagent (*Danio rerio*) | actc1b:Cavin4b-Clover | Generated in this paper | Addgene: 163854 | Construct for generating stable transgenic fish line; see Materials and methods for line generation |
| Genetic reagent (*Danio rerio*) | pT3TS-BASU-EGFP | Generated in this paper | Addgene: 163845 | Construct for in vitro RNA synthesis and RNA injection |
| Genetic reagent (*Danio rerio*) | pT3TS-TurboID-EGFP | Generated in this paper | Addgene: 163846 | Construct for in vitro RNA synthesis and RNA injection |
| Genetic reagent (*Danio rerio*) | pT3TS-miniTurbo-EGFP | Generated in this paper | Addgene: 163847 | Construct for in vitro RNA synthesis and RNA injection |
| Genetic reagent (*Danio rerio*) | pT3TS-TurboID-CaaX | Generated in this paper | Addgene: 163848 | Construct for in vitro RNA synthesis and RNA injection |
| Genetic reagent (*Danio rerio*) | pT3TS-nls-TurboID | Generated in this paper | Addgene: 163849 | Construct for in vitro RNA synthesis and RNA injection |
| Genetic reagent (*Danio rerio*) | pT3TS-CD44b-TurboID | Generated in this paper | Addgene: 163850 | Construct for in vitro RNA synthesis and RNA injection |

*Continued on next page*

*Continued*

| Reagent type (species) or resource | Designation | Source or reference | Identifiers | Additional information |
|---|---|---|---|---|
| Genetic reagent (*Danio rerio*) | pT3TS-Cavin4b-TurboID | Generated in this paper | Addgene: 163851 | Construct for in vitro RNA synthesis and RNA injection |
| Genetic reagent (*Danio rerio*) | actc1b:mRuby2-GBP | Generated in this paper | Addgene: 163856 | Construct for transient expression in zebrafish |
| Genetic reagent (*Danio rerio*) | actc1b:mRuby2-dGBP | Generated in this paper | Addgene: 163855 | Construct for transient expression in zebrafish |
| Antibody | Anti-Myc (Mouse monoclonal) | Cell Signaling Technology | 2276S | (1:2000) dilution with 5% skim milk in PBST |
| Antibody | Anti-Actin (Mouse monoclonal) | EMD Millipore | MAB1501 | (1:5000) dilution with 5% skim milk in PBST |
| Antibody | Anti-Cavin4b (Rabbit polyclonal) | Boster Biological Technology | DZ33949 | Customised antibody against zebrafish Cavin4b; (1:1000) dilution with 3% skim milk in PBST |
| Recombinant DNA reagent | p5E-actb2 | *Kwan et al., 2007*; DOI: 10.1002/dvdy.21343 | N/A | |
| Recombinant DNA reagent | p5E-actc1b | *Ariotti et al., 2018b*; DOI: 10.1242/dev.034561 | N/A | |
| Recombinant DNA reagent | pME-BASU-NS | generated in this paper | Addgene: 166565 | Gateway Entry clone contains BASU without a stop codon; see Materials and methods for cloning and Addgene for vector map |
| Recombinant DNA reagent | pME-TurboID-NS | generated in this paper | Addgene: 166566 | Gateway Entry clone contains TurboID without a stop codon; see Materials and methods for cloning and Addgene for vector map |
| Recombinant DNA reagent | pME-nls | *Ariotti et al., 2018a*; DOI: 10.1371/journal.pbio.2005473 | Addgene: 108882 | |
| Recombinant DNA reagent | pME-CD44b | *Hall et al., 2020* DOI: 10.1038/s41467-020-17486-w | Addgene: 109576 | |
| Recombinant DNA reagent | pME-miniTurbo-NS | generated in this paper | Addgene: 166567 | Gateway Entry clone contains miniTurbo without a stop codon; see Materials and methods for cloning and Addgene for vector map |
| Recombinant DNA reagent | pME-Cavin1a | *Hall et al., 2020*; DOI: 10.1038/s41467-020-17486-w | Addgene: 126927 | |
| Recombinant DNA reagent | pME-Cavin4a | *Hall et al., 2020*; DOI: 10.1038/s41467-020-17486-w | Addgene: 109562 | |
| Recombinant DNA reagent | pME-Cavin4b | *Hall et al., 2020*; DOI: 10.1038/s41467-020-17486-w | Addgene: 109563 | |
| Recombinant DNA reagent | pME-mKate2-P2A-TurboID-NS | Generated in this paper | Addgene: 166568 | Gateway Entry clone contains mKate2-P2A-TurboID without a stop codon; see Materials and methods for cloning and Addgene for vector map |
| Recombinant DNA reagent | pME-mRuby2-NS | Generated in this paper | Addgene: 166569 | Gateway Entry clone contains mRuby2 without a stop codon; see Materials and methods for cloning and Addgene for vector map |
| Recombinant DNA reagent | p3E-TurboID | Generated in this paper | Addgene: 166570 | Gateway Entry clone contains TurboID with a stop codon; see Materials and methods for cloning and Addgene for vector map |
| Recombinant DNA reagent | p3E-Clover | Generated in this paper | Addgene: 126572 | Gateway Entry clone contains Clover with a stop codon; see Materials and methods for cloning and Addgene for vector map |

*Continued on next page*

*Continued*

| Reagent type (species) or resource | Designation | Source or reference | Identifiers | Additional information |
|---|---|---|---|---|
| Recombinant DNA reagent | p3E-EGFP | Generated in this paper | Addgene: 126573 | Gateway Entry clone contains EGFP with a stop codon; see Materials and methods for cloning and Addgene for vector map |
| Recombinant DNA reagent | p3E-csGBP (dGBP) | *Ariotti et al., 2018a*; DOI: 10.1371/journal.pbio.2005473 | Addgene: 108891 | Gateway Entry clone contains csGBP with a stop codon; see Materials and methods for cloning and Addgene for vector map |
| Recombinant DNA reagent | p3E-GBP | *Ariotti et al., 2015a*; DOI: 10.1016/j.devcel.2015.10.016 | Addgene: 67672 | Gateway Entry clone contains GBP with a stop codon; see Materials and methods for cloning and Addgene for vector map |
| Recombinant DNA reagent | p3E-CaaX (tH) | *Hall et al., 2020*; DOI: 10.1038/s41467-020-17486-w | Addgene: 109539 | |
| Recombinant DNA reagent | pT3TS-DEST | Generated in this paper | Addgene: 166571 | Gateway Destination vector contains T3 and T7 promoters for in vitro RNA synthesis; see Materials and methods for cloning and Addgene for vector map |
| Peptide, recombinant protein | Streptavidin-HRP | Abcam | Ab7403 | (1:5000) dilution with 5% BSA in PBST |
| Peptide, recombinant protein | Proteinase K | Invitrogen | 25530015 | |
| Peptide, recombinant protein | Pronase | Roche | 10165921001 | |
| Peptide, recombinant protein | Trypsin/Lys-C Mix, Mass Spec Grade | Promega | V5073 | |
| Commercial assay or kit | Pierce BCA protein assay kit | Thermo Scientific | 23225 | |
| Commercial assay or kit | Clarity Western ECL Substrate | Bio-Rad | 1705061 | |
| Commercial assay or kit | InstantBlue | Expedeon | ISB1L-1L | |
| Commercial assay or kit | SYPRO Ruby Protein Gel Stain | Invitrogen | S12000 | |
| Chemical compound, drug | Biotin | Sigma-Aldrich | B4639-1G | |
| Chemical compound, drug | Phenol Red | Sigma-Aldrich | P0290-100ML | |
| Chemical compound, drug | NeutrAvidin-DyLight 405 | Invitrogen | 22831 | |
| Chemical compound, drug | NeutrAvidin-DyLight 550 | Invitrogen | 84606 | |
| Chemical compound, drug | Sodium deoxycholate | Sigma-Aldrich | D6750-10G | |
| Chemical compound, drug | NP-40 | Sigma-Aldrich | 18896–50 ML | |
| Chemical compound, drug | EDTA | Astral Scientific | BIOEB0185-500G | |
| Chemical compound, drug | Complete Protease Inhibitor Cocktail | Sigma-Aldrich | 11836145001 | |
| Chemical compound, drug | Paraformaldehyde | Sigma-Aldrich | P6148-500G | |

*Continued on next page*

*Continued*

| Reagent type (species) or resource | Designation | Source or reference | Identifiers | Additional information |
|---|---|---|---|---|
| Chemical compound, drug | PBS tablets | Medicago | 09-8912-100 | |
| Chemical compound, drug | Triton-X100 | Sigma-Aldrich | T9284-500ML | |
| Chemical compound, drug | Tween 20 | Sigma-Aldrich | P1379-500ML | |
| Chemical compound, drug | DAPI | Sigma-Aldrich | D9542-5MG | |
| Chemical compound, drug | Bolt LDS sample buffer (4X) | Invitrogen | B0008 | |
| Chemical compound, drug | Dynabeads MyOne Streptavidin C1 | Invitrogen | 65001 | |
| Chemical compound, drug | Agarose, low gelling temperature | Sigma-Aldrich | A9414-100G | |
| Software, algorithm | ProteinPilot | SCIEX | | Version 5.0.1 |
| Software, algorithm | Analyst TF | SCIEX | | Version 1.7 |
| Software, algorithm | Excel | Microsoft | | Version 16.45 |
| Software, algorithm | Prism8 | GraphPad | | Version 8.0.2 |
| Software, algorithm | Fiji | ImageJ | | Version 2.0.0-rc-69/1.52 p |
| Software, algorithm | Illustrator | Adobe | | Version 23.1.1 |
| Other | PD-10 desalting column | GE Healthcare | 17-0851-01 | |
| Other | LoBind tube | Eppendorf | 022431048 | |
| Other | Blot 4–12% Bis-Tris Plus precast gels | Invitrogen | NW04120BOX | |

## Zebrafish husbandry

Zebrafish were raised and maintained according to institutional guidelines (Techniplast recirculating system, 14 hr light/10 hr dark cycle, University of Queensland, UQ). Adults (90 dpf above) were housed in 3 or 8 L tanks with flow at 28.5°C and embryos up to five dpf were housed in 8 cm Petri dishes in standard E3 media (5 mM NaCl, 0.17 mM KCl, 0.33 mM $CaCl_2$, and $MgSO_4$) at 28°C (incubated in the dark) (*Westerfield, 2007*). All experiments were approved by the University of Queensland Animal Ethics Committee. The following zebrafish strains were used in this study: wild-type (TAB), an AB/TU line generated in UQBR Aquatics (UQ Biological Resources), *Tg(actc1b:cavin1a-Clover)*, *Tg(actc1b:cavin4a-Clover)*, *Tg(actc1b:Cavin4b-Clover)*, *Tg(actb2:mKate2-P2A-TurboID-dGBP)*, *Tg(kdrl:eGFP)* and *Tg(MotoN:GFP)*. The developmental stages of zebrafish used in experiments are prior to specific sex determination. All zebrafish used in experiment were healthy, not involved in previous procedures and drug or test naive.

## DNA constructs and transgenic fish lines

The protein sequence of TurboID and MiniTurbo was constructed according to *Branon et al., 2018* while the protein sequence of BASU was designed according to *Ramanathan et al., 2018*. The coding sequences of TurboID, MiniTurbo, and BASU were ordered from IDT as gene fragment with codon optimised for zebrafish expression (https://sg.idtdna.com). The expression of biotin ligases was driven by a ubiquitous promoter of *actb2* (*Higashijima et al., 1997*; *Casadei et al., 2011*). A red fluorescent reporter, mKate2, was indirectly linked into the N-terminus of biotin ligase through a self-cleaving P2A sequence (*Shcherbo et al., 2009*; *Kim et al., 2011*). Promoter, fluorescent report and biotin ligase were cloned into destination vector using Gateway cloning system. All fish lines were generated by using Tol2kit system according to established methods (*Kawakami, 2004*; *Kwan et al., 2007*). In brief, plasmid constructs for generating transgenic lines were co-injected with

tol2 mRNA into one-cell-stage WT zebrafish embryos (*Nusslein-Volhard and Dahm, 2002*). Injected $F_0$s were raised and screened for founders producing positive $F_1$s with Mendelian frequencies, indicative of single genomic integration. Positive $F_1$s grown to reproductive age were used for our experiments. Stable lines were maintained as heterozygotes. All stable lines used are given in the Key Resource Table.

## Transient expression by DNA/RNA microinjection

DNA plasmid and RNA transcript for injection were diluted to final concentration of 30 ng/µl and 200 ng/µl, respectively, with addition of Phenol Red (Sigma-Aldrich) as injection tracer. A bolus of approximately 1/5 of the total cell diameter was injected into each embryo. For DNA injection, the bolus was injected into the cell of embryos at single cell stage (5–25 min-post-fertilisation). For RNA injection, the bolus was injected into the yolk of the embryos up until two-cell stage. The RNA transcript was synthesised by mMESSAGE mMACHINE T3 (Invitrogen) according to manufacturer's instruction. The RNA transcripts were tagged with poly(A) tail using Poly(A) Tailing Kit (Invitrogen) to extend the stability of mRNA in zebrafish embryos.

## In vivo biotin labelling

Embryos at indicated developmental stage were incubated in the E3 media supplemented with 500 µM biotin for 18 hr to initiate biotinylation in vivo. For embryos before hatching, a dechorionation step was carried out by using Pronase (Roche, 100 µg/ml in E3 media for 40 min at 28°C) prior to the biotin incubation. After biotin incubation, embryos were washed for 40 min with two changes of standard E3 media to remove unincorporated biotin before subsequent immunostaining or protein extraction.

## Zebrafish embryos protein extraction

Fish embryos after in vivo biotin labelling were deyolked by mechanical disruption through a 200 µl pipette tips in calcium-free Ringer's solution followed by two changes of solution at 4°C. The deyolked embryos was lysed by brief sonication in RIPA buffer (50 mM Tris-HCl, pH 7.5; 150 mM NaCl; 1% NP-40; 0.1% SDS; 5 mM EDTA; 0.5% Na-deoxycholate,) with freshly added cOmplele Protease Inhibitor. Lysates were further solubilised at 4°C with rotation for 30 min. Insoluble material was removed by centrifugation at 14,000 ×g for 10 min at 4°C, and supernatant were collected for BCA protein assay determining protein concentration. For western blot analysis, 25 fish embryos per group were used for protein extraction, whilst, for streptavidin affinity purification, approximately 350 embryos were used for each group.

## Western blotting

Western blot analysis was performed largely as described previously *Lo et al., 2015*. Briefly, zebrafish samples from protein extraction were mixed with NuPAGE LDS sample buffer (4X) and 10 mM DTT. Protein samples were analysed by Western blotting with following antibodies: mouse anti-Myc (dilution 1:2000), mouse anti-Actin (dilution 1:5000), rabbit anti-Cavin4b (dilution 1:2000), anti-mouse and anti-rabbit HRP-conjugated antibodies (dilution 1:5000), streptavidin-HRP (dilution 1:5000). ECL blotting reagent was used to visualise HRP and chemiluminescent signal was detected using the ChemiDoc MP system (BioRad) as per the manufacture's instruction.

## Streptavidin beads pulldowns

Fresh embryo protein extracts (4 mg in 2.5 ml RIPA buffer) was passed through PD-10 desalting column (GE Healthcare) to remove excess free biotin using the gravity protocol according to manufacturer's instruction. Protein extracts were then mixed with Dynabeads MyOne Streptavidin C1 (Invitrogen) from 200 µl bead slurry that were pre-washed with RIPA buffer, and incubated on a rotor wheel at 4°C overnight (16 hr). The next day, the beads were separated from the protein extracts on a magnetic rack and transferred to a new 2 ml LoBind tube (Eppendorf). The beads were washed with 1 ml of each following solution: twice with RIPA buffer, once with 2% SDS in 50 mM Tris-HCl pH7.5, once with 2 M urea in 10 mM Tris-HCl pH8.0 and twice again in RIPA buffer without cOmplete Protease Inhibitor. Washed beads were boiled in 60 µl of 2X Blot LDS sample buffer (4X diluted to 2X with RIPA buffer) containing 2 mM biotin and 20 mM DTT at 95°C for 10 min with 10 s

vortex after first 5 min boiling. Five ul μof the pulldown samples was used for immunoblots, whereas the 50 μl of the samples were used for SDS-PAGE with SYPRO Ruby (Invitrogen) or InstantBlue protein gel stain.

## Immunostaining and confocal microscopy

Fish embryos after in vivo biotin labelling were fixed in 4% paraformaldehyde (PFA) overnight at 4°C. After fixation, embryos were permeabilised by proteinase K (10 μg/ml, 10 min for embryos at 2 dpf, or 15 min for embryos at 3 dpf), and fixed again with 4% PFA for 15 min. Embryos were washes with PBS-Tween 20 (0.1%) and blocked in PBS with 0.3% Triton X-100% and 4% BSA for 3 hr at room temperature. Staining was performed in blocking buffer with NeutrAvidin-DyLight (1:500 dilution) overnight at 4°C followed by four washes with PBS 0.3% Trition X-100. For nuclear staining, embryos were stained with DAPI for 10 min followed by three washes with PBS 0.3% Trition X-100.

Confocal imaging was performed on Zeiss 710 meta upright confocal microscopes. Zebrafish embryos were mounted in 1% low melting point agarose in embryos media (*Westerfield, 2007*) on a standard 8 cm petri dish. Objectives used were Zeiss water immersion x40 N/A 1.0 (catalogue number 420762). For live embryo imaging, embryos were anaesthetised in 2.5 mM tricaine prior to imaging.

## Sample preparation for mass spectrometry

For in-gel digestion, the streptavidin pulldown samples were separated by SDS-PAGE on a 4–12% precast gel (Blot Bis-Tris Plus, Invitrogen) and then stained with Instant*Blue* (Expedeon). The whole lane was excised from the gel and future cut into approximate $3 \times 1 \times 2$ mm$^3$ slices (L x W x H) and each slice were placed into a separate LoBind tubes (Eppendorf) for destaining. The gels were destained by adding 500 μl of 100 mM ammonium bicarbonate/acetonitrile (1:1, vol/vol) and incubated with occasional vortexing for 30 min. The ammonium bicarbonate/acetonitrile buffer was removed, and the gel pieces dried for 15 min by the addition of 200 μl of acetonitrile. The acetonitrile was removed and another 200 μl aliquot was added and left for 15–30 min. The acetonitrile was removed in preparation for trypsin digestion.

After destain, the gel pieces were covered with 200 μl of 20 ng/μl of sequence grade trypsin/Lys-C (Promega) in 50 mM ammonium bicarbonate pH8 buffer. The gel pieces were left for 1 hr and if required a further 100 μl of trypsin/Lys-C solution was added to cover the gel pieces. The samples were placed in an incubator at 37°C overnight. The trypsin solution was transferred from each sample and placed in a clean Eppendorf tube. Of 5% formic acid/acetonitrile (3:1, vol/vol), 200 μl was added to each tube and incubated for 15 min at room temperature in a shaker. The supernatant was placed into the pre-cleaned Eppendorf tubes, together with the trypsin solution for each sample and dried down in a vacuum centrifuge.

For HPLC/MS MS/MS analysis, 12 μl of 1.0% (vol/vol) trifluoroacetic acid in water was added to the tube, which was vortexed and incubated for 2 min in the sonication bath and then centrifuged for 1 min at 6700 xg (10,000 rpm) and finally, transferred to an autosampler vial for analysis.

## Lipid chromatography and mass spectrometry

The tryptic peptide extracts were analysed by nanoHPLC/MS MS/MS on an Eksigent ekspert nanoLC 400 system (SCIEX) coupled to a Triple TOF 6600 mass spectrometer (SCIEX) equipped with Pico-View nanoflow (New Objective) ion source. Five μl of each extract was injected onto a 5 mm x 300 μm, C18 3 μm trap column (SGE, Australia) for 5 min at 10 μl/min. The trapped tryptic peptide extracts were then washed onto the analytical 75 μm x 150 mm ChromXP C18 CL 3 μm column (SCIEX) at 400 nl/min and a column temperature of 45°C. Linear gradients of 2–40% solvent B over 60 min at 400 nl/min flowrate, followed by a steeper gradient from 40% to 90% solvent B in 5 min, then 90% solvent B for 5 min, were used for peptide elution. The gradient was then returned to 2% solvent B for equilibration prior to the next sample injection. Solvent A consisted of 0.1% formic acid in water and solvent B contained 0.1% formic acid in acetonitrile. The ionspray voltage was set to 2600V, declustering potential (DP) 80V, curtain gas flow 30, nebuliser gas 1 (GS1) 30, interface heater at 150°C. The mass spectrometer acquired 50 ms full scan TOF-MS data followed by up to 30 100 ms full scan product ion data, with a rolling collision energy, in an Information-Dependant Acquisition, IDA, mode for protein identification and peptide library production. Full scan TOFMS data

was acquired over the mass range 350–1800 and for product ion ms/ms 100–1500. Ions observed in the TOF-MS scan exceeding a threshold of 200 counts and a charge state of +two to +five were set to trigger the acquisition of product ion, ms/ms spectra of the resultant 30 most intense ions.

### MS data analysis and GO annotation

Data was acquired and processed using Analyst TF 1.7 software (SCIEX). Protein identification was carried out using ProteinPilot software v5.0 (SCIEX) with Paragon database search algorithm. MS/MS spectra were searched against the zebrafish proteome in the UniProt database (2019 version, containing 46847 proteins and 28365 common contaminants). The search parameter was set to through with False Discovery Rate (FDR) analysis. A non-linear fitting method was used to determine both a global and a local FDR from the decoy database search (*Tang et al., 2008*). The cut-off for identified proteins was set to 1% global FDR. Endogenous biotinylated proteins, common contaminants and background proteins (generated from TurboID-dGBP control embryos) were subtracted from the cavin proteomes. The MS2Count was calculated for each identified protein by summing the MS2Count of all peptides belonging to that protein. The MS2Count for each protein was used to generate heatmaps for semiquantitative comparison across cavin interactome.

The GO annotation for our identified proteins was carried out using DAVID 6.8 (*Huang et al., 2009b*; *Huang et al., 2009a*). For zebrafish proteins without annotation information, manually annotation was performed by searching their human or mouse homologs.

## Acknowledgements

We thank Dr. Anne Lagendijk and Dr. Jean Giacomotto for *Tg(kdrl:EGFP)* and *Tg(MotoN:EGFP)* fish lines, respectively. Confocal microscopy was performed at the Australian Cancer Research Foundation (ACRF)/Institute for Molecular Bioscience (IMB) Dynamic Imaging Facility for Cancer Biology with funding from the ACRF. MS spectrometry was performed at the IMB Mass Spectrometry Facility, the University of Queensland. This work was supported by fellowship and grants from the National Health and Medical Research Council of Australia (to RG P, grant numbers 569542 and 1045092; to RGP, grant number APP1044041; and to TE H and RGP, grant number APP1099251), the Australian Research Council (to TEH and PGP, grant number DP200102559) as well as by the Australian Research Council Centre of Excellence in Convergent Bio-Nano Science and Technology (to RGP, grant number CE140100036).

## Additional information

### Funding

| Funder | Grant reference number | Author |
| --- | --- | --- |
| National Health and Medical Research Council | 569542 | Robert G Parton |
| National Health and Medical Research Council | 1045092 | Robert G Parton |
| National Health and Medical Research Council | APP1044041 | Robert G Parton |
| National Health and Medical Research Council | APP1099251 | Robert G Parton Thomas E Hall |
| Australian Research Council | DP200102559 | Robert G Parton Thomas E Hall |
| Australian Research Council | CE140100036 | Robert G Parton |

The funders had no role in study design, data collection and interpretation, or the decision to submit the work for publication.

## Author contributions
Zherui Xiong, Conceptualization, Data curation, Formal analysis, Validation, Investigation, Visualization, Methodology, Writing - original draft, Project administration; Harriet P Lo, Resources, Supervision, Methodology, Writing - review and editing; Kerrie-Ann McMahon, Supervision, Methodology, Writing - review and editing; Nick Martel, Resources, Methodology; Alun Jones, Software, Investigation; Michelle M Hill, Writing - review and editing; Robert G Parton, Conceptualization, Resources, Supervision, Funding acquisition, Methodology, Writing - review and editing; Thomas E Hall, Conceptualization, Supervision, Funding acquisition, Investigation, Methodology, Writing - review and editing

## Author ORCIDs
Zherui Xiong (iD) http://orcid.org/0000-0002-5038-5629
Michelle M Hill (iD) http://orcid.org/0000-0003-1134-0951
Robert G Parton (iD) https://orcid.org/0000-0002-7494-5248
Thomas E Hall (iD) https://orcid.org/0000-0002-7718-7614

## Ethics
Animal experimentation: This study was approved by Institutional Biosafety Committee, Office of the Gene Technology Regulator, Australian Government Department of Health, and Molecular Biosciences Animal Ethics Committees, the University of Queensland. The ethics approval numbers are IMB/271/19/BREED and IMB/326/17. The IBC/OGTR approval number is IBC/1080/IMB/2017.

## Decision letter and Author response
Decision letter https://doi.org/10.7554/eLife.64631.sa1
Author response https://doi.org/10.7554/eLife.64631.sa2

# Additional files
### Supplementary files
• Supplementary file 1. Tables showing all identified and enriched proteins in Cavin1a (Table 1), Cavin4a (Table 2), and Cavin4b (Table 3) samples as well as *localisation and functions* annotation (Table 4).

• Supplementary file 2. Table showing all the peptides identified in Cavin1a and sibiling control samples.

• Supplementary file 3. Table showing proteins identified in control zebrafish embryos expressing only TurboID-dGBP.

• Transparent reporting form

### Data availability
All data generated or analysed during this study are included in the manuscript and supplementary files. Source data files have been provided for Figures 1, 2, 4 and 5.

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
