## [Decision Letter]

**Acceptance summary:**

This paper reports an important new advance in mapping protein-protein proximity networks, and as such will be of interest to scientists across biological disciplines. In this modular system, a conditionally stabilized GFP-binding nanobody expressed as a transgene can be combined with other zebrafish transgenic lines expressing GFP-tagged proteins of interest to achieve biotin-dependent proteomic profiling. The data show convincingly in vivo labeling and identification of proximal proteins within a living organism. This work significantly expands experimental toolbox for defining protein interaction networks in zebrafish and beyond.

**Decision letter after peer review:**

Thank you for submitting your article "in vivo proteomic mapping through GFP-directed proximity-dependent biotin labelling in zebrafish" for consideration by *eLife*. Your article has been reviewed by three peer reviewers, including Lilianna Solnica-Krezel as the Reviewing Editor and Reviewer #1, and the evaluation has been overseen by Richard White as the Senior Editor. The following individual involved in review of your submission has agreed to reveal their identity: Ben M Major (Reviewer #3).

The reviewers have discussed the reviews with one another and the Reviewing Editor has drafted this decision to help you prepare a revised submission.

Essential revisions:

This is a very well written manuscript detailing an important technical advance for the protein-protein interaction community. The goal of the study was to develop the BLITZ method, which was accomplished and is now established. Comments and questions of minor or technical impact are as follows:

The authors conclude that "TurboID exhibited the strongest biotinylation of endogenous proteins among the three new biotin ligases" when transiently expressed in early zebrafish embryos (Figure 1B). However, Figure 1B also shows that by western blotting, TurboID was expressed at higher levels. Were biotinylation results normalized to the ligase expression levels?

Randomly transgenes inserted into the zebrafish genome can be silenced in consecutive generations. Can the authors comment whether TurboID-dGBP protein expression remains stable over generations? Have attempts been made to insert the TurboID-dGBP under the control of endogenous, possibly tissue-specific promoters. Some background biotinylation has been observed even with dGBP, and tissue specific expression would help reduce it.

In Figure 4A, it is unclear why Cavlin1a-Clover shows co-localization with NeutrAvidin in the third row from the top but not in the second row from the bottom. The authors suggest that the biotinylation seen by Western blot in Figure 4B in the absence of Cavlin1a is driven by dGBP en route to the proteasome. Given the importance of this control from an in vivo setting, further definition and understanding of this signal and how it impacts true positive discovery is needed.

Overall, the statistical analysis of significance for the scored proximal proteins is weak and lacks a more robust and field standard approach. Specifically, a nice set of control transgenic lines have been developed, but have not been processed by MS and are thus not used in scoring. Data processing describes a normalization step to endogenously expressed carboxylases. Do these carboxylases change in abundance across different bait proteins? The MS acquisition parameters are lacking from the Materials and methods section.

The endogenously biotinylated protein they used in their analysis was propionyl-CoA carboxiylase subunit alpha, a mitochondrial protein. Given that this is a proximity label, it might be good to add some additional controls used in their analyses, for protein specificity that are not restricted to mitochondria. If they compare the mass spec identified profile of NLS-TurboID, cytoplasmic TurboID GFP and CaaX TurboID GFP to the Caveolins. Some more statements of the restrictions of biotinylation as a proximity/location label would be appreciated in the Discussion.

The experiments describing the use of the TurboID-dGBP system to map proteins associated with Cavin9 and 4 in zebrafish muscle are very informative. With this experiment showing the potential of the method, it would be important to validate some of the newly identified potential interactors.

Only weak biotin labeling was detected after 4-6 h incubation of zebrafish embryos with biotin, and adequate labeling required 18 h incubation. Is this related to low permeability of zebrafish embryos to biotin? Has injection of biotin into embryos been attempted? This would not address later stages of development, but could be beneficial during embryogenesis.

---

## [Author Response]

Essential revisions:This is a very well written manuscript detailing an important technical advance for the protein-protein interaction community. The goal of the study was to develop the BLITZ method, which was accomplished and is now established. Comments and questions of minor or technical impact are as follows:The authors conclude that "TurboID exhibited the strongest biotinylation of endogenous proteins among the three new biotin ligases" when transiently expressed in early zebrafish embryos (Figure 1B). However, Figure 1B also shows that by western blotting, TurboID was expressed at higher levels. Were biotinylation results normalized to the ligase expression levels?

The biotinylation level was not normalised to the ligase expression level. In our hands, we found that both BASU and miniTurbo were expressed at a lower level than TurboID when an equal amount of RNA was injected into zebrafish embryos. While we did not explore the underlying reasons for this, greater instability of miniTurbo has been reported previously (May et al., 2020) and other biotin ligases have shown poor expression in cell cultures (Branon et al., 2018). Regardless of mechanism, BASU and miniTurbo did not perform as well as TurboID in our system. These points have been clarified in the Results section as follows:

“Of note, BASU and miniTurbo were expressed at a lower level than TurboID despite equal amount of RNA injection. While we did not explore the underlying reasons for this, greater instability of miniTurbo has been reported previously (May et al., 2020) and other biotin ligases have shown poor expression in cell cultures (Branon et al., 2018).”

Randomly transgenes inserted into the zebrafish genome can be silenced in consecutive generations. Can the authors comment whether TurboID-dGBP protein expression remains stable over generations? Have attempts been made to insert the TurboID-dGBP under the control of endogenous, possibly tissue-specific promoters. Some background biotinylation has been observed even with dGBP, and tissue specific expression would help reduce it.

When generating stable lines in our laboratory we take great care to avoid the problems of epigenetic silencing and loss of transgene activity due to dilution of multiple genomic integrations. Before selecting a line to work with, we outcross over several generations to ensure firstly that the transgene is inherited in a Mendelian fashion, and secondly to ensure that we have selected lines that are not prone to silencing. In generating the lines for this study, we observed that the expression of TurboID-dGBP transgene remained stable over four generations and continues to remain so. This point has been expanded upon as follows:

“The expression of transgene is stable in our zebrafish lines and demonstrates Mendelian inheritance over four generations, indicating a stable single transgenic integration.”

We did not attempt to insert TurboID-dGBP under the control of endogenous promoter. Inserting the biotin ligases at endogenous loci is theoretically possible, but the great utility of our dGBP mediated approach is that it can be used with any GFP-expressing line. This means that a more flexible and streamlined approach would be to make or source GFP knock-in fish to cross with our dGBP line. We have used tissue specific expression of TurboID-dGBP to conduct proteomic experiments. However, we were unable to ascertain whether there was any significant reduction in background biotinylation since background biotinylation is already very low. Our rationale has been that the use of tissue specific promoters would require the generation of new driver lines for each application, decreasing the utility of the system. However, such an approach could conceivably be useful where GFP expression from the reporter line is expressed in two or more different tissues or cell types.

In Figure 4A, it is unclear why Cavlin1a-Clover shows co-localization with NeutrAvidin in the third row from the top but not in the second row from the bottom. The authors suggest that the biotinylation seen by Western blot in Figure 4B in the absence of Cavlin1a is driven by dGBP en route to the proteasome. Given the importance of this control from an in vivo setting, further definition and understanding of this signal and how it impacts true positive discovery is needed.

We believe this misunderstanding has arisen because of incomplete labelling of this figure, and we have amended appropriately. The second row from the bottom in Figure 4A is a control group consisting sibling embryos carrying only Cavin1a-Clover but *not* the TurboID-dGBP transgene (the presence of the TurboID transgene is shown by the mKate2 signal in the third column). Additional labels “GFP-negative; mKate2-negative” have been added to Figure 4A and to the figure legend for clarification.

Importantly, mass-spec analysis has been performed on the negative control group expressing TurboID-dGBP without GFP bait protein and the data are given in Supplementary file 3. The majority are endogenous biotinylated proteins, pyruvate carboxylase a/b (pcxa/b), Acetyl-CoA Carboxylase beta (acacb) and Propionyl-CoA carboxylase (pcc). In addition, several nuclear proteins (ap3d and taf3), ribosomal proteins (rpl37, rpl36a and rpl39), cytoskeletal proteins (krt5 and myhz1.1) and zebrafish yolk proteins (vitellogenins) were also identified. These proteins are commonly found in BioID-based MS sequencing and were excluded from analysis. This point has been clarified further as follows:

“Subsequent MS analysis revealed these background proteins are mainly endogenous biotinylated proteins, nuclear proteins, cytoskeletal proteins and yolk proteins (Supplementary file 3).”

Overall, the statistical analysis of significance for the scored proximal proteins is weak and lacks a more robust and field standard approach. Specifically, a nice set of control transgenic lines have been developed, but have not been processed by MS and are thus not used in scoring. Data processing describes a normalization step to endogenously expressed carboxylases. Do these carboxylases change in abundance across different bait proteins? The MS acquisition parameters are lacking from the Materials and methods section.

Our main objective for the mass spec analysis was to provide proof of principle at molecular level of the specificity and accuracy of the BLITZ system. For this reason, we chose a qualitative rather than quantitative approach. However, this comment prompted us to revisit our analysis methodology. We have rerun the protein identification search using ProteinPilot software with false discovery rate (FDR) set at 1% global FDR. The FDR report detailing the quality of protein and peptide identification is now include as Figure 5—source data 1-6. Label-free quantitation used summed MS2Count for each protein. Non-specific proteins identified in pulldown of sibling embryos that express only the TurboID-dGBP transgene was used as background control. The following explanation has been added:

“The sibling embryos inheriting only the TurboID transgene were used as a background control (Figure 5A). […] Individual hits and general properties of putative interacting proteins are discussed below.”

Based on our Western Blot and mass spec analysis, the abundance of endogenous biotinylated proteins stayed at a similar level across different fish lines used.

The mass spec acquisition parameters are now stated in the Materials and methods section follows:

“Data was acquired and processed using Analyst TF 1.7 software (SCIEX). Protein identification was carried out using ProteinPilot software v5.0 (SCIEX) with Paragon database search algorithm. […] The MS2Count for each protein was used to generate heatmaps for semiquantitative comparison across cavin interactome”

The endogenously biotinylated protein they used in their analysis was propionyl-CoA carboxiylase subunit alpha, a mitochondrial protein. Given that this is a proximity label, it might be good to add some additional controls used in their analyses, for protein specificity that are not restricted to mitochondria. If they compare the mass spec identified profile of NLS-TurboID, cytoplasmic TurboID GFP and CaaX TurboID GFP to the Caveolins.

In our original submission propionyl-CoA carboxylase subunit alpha was used for normalisation of the Protein Score. In this revised manuscript, we have now used MS2Count for relative label-free quantification of detected proteins across the samples. The controls were the respective sibling embryos as explained above and depicted in Figure 5A. We did not generate stable transgenic fish lines expressing NLS-TurboID, cytoplasmic TurboID-GFP or TurboID-CaaX. The fish embryos expressing these TurboID variants (Figure 2) were generated using direct injection of RNA so it is not feasible to conduct mass spec analysis on these animals. However, we believe our existing mass spec data is sufficient to demonstrate the specificity and accuracy of BLITZ method.

Some more statements of the restrictions of biotinylation as a proximity/location label would be appreciated in the Discussion.

We have added the following text to the Discussion:

“As BLITZ uses biotin as a label, the method (like most BioID methods) is problematic for the identification of interactors that are endogenously biotinylated, such as carboxylases. […] Thus, subsequent validation using other independent approaches such as biomolecular fluorescence complementation and affinity pulldown is essential to distinguish bona fide interactors from non-interacting neighbouring proteins.”

The experiments describing the use of the TurboID-dGBP system to map proteins associated with Cavin9 and 4 in zebrafish muscle are very informative. With this experiment showing the potential of the method, it would be important to validate some of the newly identified potential interactors.

We have validated one of the candidates, Bin1b, as a bona fide interactor of Cavin4 (Lo et al., unpublished, https://www.biorxiv.org/content/10.1101/2021.01.13.426456v1). A systematic validation of all candidates is in progress and forms the basis of another study.

Only weak biotin labeling was detected after 4-6 h incubation of zebrafish embryos with biotin, and adequate labeling required 18 h incubation. Is this related to low permeability of zebrafish embryos to biotin? Has injection of biotin into embryos been attempted? This would not address later stages of development, but could be beneficial during embryogenesis.

We have not attempted biotin injection into zebrafish embryos for the same reasons we do not use RNA injections for mass spec analyses. We require approximately 400 embryos for each sample group which makes injection an unfeasible option. However, injection of biotin solution into zebrafish has recently been applied successfully to adult zebrafish which yield more tissue (Pronobis et al., unpublished, https://www.biorxiv.org/content/10.1101/2021.01.19.427246v1?ct=).